# Async-RED: A Provably Convergent Asynchronous Block Parallel Stochastic Method using Deep Denoising Priors

**Yu Sun[1], Jiaming Liu[1], Yiran Sun[1], Brendt Wohlberg[2], Ulugbek S. Kamilov[1]**
[1]Washington University in St. Louis, [2]Los Alamos National Laboratory
{sun.yu,jiaming.liu,yiran.s,kamilov}@wustl.edu, brendt@ieee.org

## Abstract

Regularization by denoising (RED) is a recently developed framework for solving inverse problems by integrating advanced denoisers as image priors. Recent work has shown its state-of-the-art performance when combined with pre-trained *deep denoisers*. However, current RED algorithms are inadequate for parallel processing on multicore systems. We address this issue by proposing a new *asynchronous RED* (Async-RED) algorithm that enables asynchronous parallel processing of data, making it significantly faster than its serial counterparts for large-scale inverse problems. The computational complexity of Async-RED is further reduced by using a random subset of measurements at every iteration. We present complete theoretical analysis of the algorithm by establishing its convergence under explicit assumptions on the data-fidelity and the denoiser. We validate Async-RED on image recovery using pre-trained deep denoisers as priors.

## 1 Introduction

Imaging inverse problems seek to recover an unknown image $x \in \mathbb{R}^n$ from its noisy measurements $y \in \mathbb{R}^m$. Such problems arise in many fields, ranging from low-level computer vision to biomedical imaging. Since many imaging inverse problems are ill-posed, it is common to regularize the solution by using prior information on the unknown image. Widely-adopted image priors include total variation, low-rank penalties, and transform-domain sparsity (Rudin et al., 1992; Figueiredo & Nowak, 2001; 2003; Hu et al., 2012; Elad & Aharon, 2006).

There has been considerable recent interest in *plug-and-play priors (PnP)* (Venkatakrishnan et al., 2013; Sreehari et al., 2016) and *regularization by denoising (RED)* (Romano et al., 2017), as frameworks for exploiting image denoisers as priors for image recovery. The popularity of deep learning has led to a wide adoption of *deep denoisers* within PnP/RED, leading to their state-of-the-art performance in a variety of applications, including image restoration (Mataev et al., 2019), phase retrieval (Metzler et al., 2018), and tomographic imaging (Wu et al., 2020). Their empirical success has also prompted a follow-up theoretical work clarifying the existence of explicit regularizers (Reehorst & Schniter, 2019), providing new interpretations based on fixed-point projections (Cohen et al., 2020), and analyzing their coordinate/online variants (Sun et al., 2019a; Wu et al., 2020). Nonetheless, current PnP/RED algorithms are inherently *serial*. As illustrated in Fig. 1, this makes them suboptimal on multicore systems that are often required for processing large-scale datasets (Recht et al., 2011), such as those involving biomedical (Ong et al., 2020) and astronomical images (Akiyama et al., 2019)

We address this gap by proposing a novel *asynchronous RED* (Async-RED) algorithm. The algorithm decomposes the inference problem into a sequence of partial (block-coordinate) updates on $x$ executed *asynchronously* in parallel over a multicore system. Async-RED leads to a more efficient usage of available cores by avoiding synchronization of partial updates. Async-RED is also scalable in terms of the number of measurements, since it processes only a small random subset of $y$ at every iteration. We present two new theoretical results on the convergence of Async-RED based on a unified set of explicit assumptions on the data-fidelity and the denoiser. Specifically, we establish its fixed-point convergence in the *batch* setting and extend this analysis to the randomized *minibatch* scenario. Our results extend recent work on *serial* block-coordinate RED (BC-RED) (Sun et al.,

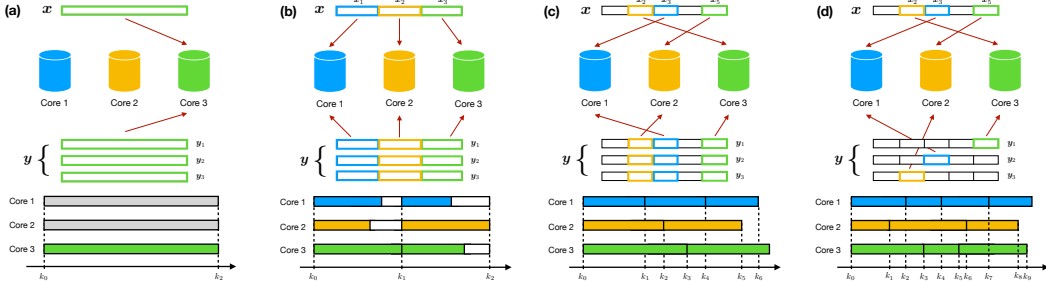

Figure 1: Visual illustration of *serial* and *parallel* image recovery on a multicore system. **(a)** Serial processing uses only one core of the system for every iteration. **(b)** Synchronous parallel processing has to wait for the slowest core to finish before starting the next iteration. **(c)** Asynchronous parallel processing can continuously iterate using all the cores without waiting. **(d)** Asynchronous parallel processing using the stochastic gradient leads to additional flexibility. **(a)**, **(b)**, and **(c)** use all the corresponding measurements at every iteration, while **(d)** uses only a small random subset at a time. ASYNC-RED adopts the schemes shown in **(c)** and **(d)**.

2019a) and are fully consistent with the traditional asynchronous parallel optimization methods (Lian et al., 2015; Sun et al., 2017). We numerically validate ASYNC-RED on image recovery from linear and noisy measurements using pre-trained deep denoisers as image priors.

## 2 BACKGROUND

**Inverse problems.** Inverse problems are traditionally formulated as a composite optimization problem

$$\widehat{\boldsymbol{x}} = \arg\min_{\boldsymbol{x} \in \mathbb{R}^n} g(\boldsymbol{x}) + h(\boldsymbol{x}), \tag{1}$$

where $g$ is the data-fidelity term that ensures consistency of $\boldsymbol{x}$ with the measured data $\boldsymbol{y}$ and $h$ is the regularizer that infuses the prior knowledge on $\boldsymbol{x}$. For example, consider the smooth $\ell_2$-norm data-fidelity term $g(\boldsymbol{x}) = \|\boldsymbol{y} - \boldsymbol{Ax}\|_2^2$, which assumes a linear observation model $\boldsymbol{y} = \boldsymbol{Ax} + \boldsymbol{e}$, and the nonsmooth TV regularizer $h(\boldsymbol{x}) = \tau\|\boldsymbol{Dx}\|_1$, where $\tau > 0$ is the regularization parameter and $\boldsymbol{D}$ is the image gradient (Rudin et al., 1992).

**Regularization by denoising (RED).** RED is a recent methodology for imaging inverse problems that seeks vectors $\boldsymbol{x}^* \in \mathbb{R}^n$ satisfying

$$\mathsf{G}(\boldsymbol{x}^*) = \nabla g(\boldsymbol{x}^*) + \tau(\boldsymbol{x}^* - \mathsf{D}_\sigma(\boldsymbol{x}^*)) = 0 \quad \Leftrightarrow \quad \boldsymbol{x}^* \in \mathsf{zer}(\mathsf{G}) := \{\boldsymbol{x} \in \mathbb{R}^n : \mathsf{G}(\boldsymbol{x}) = 0\} \quad (2)$$

where $\nabla g$ denotes the gradient of the data-fidelity term and $\mathsf{D}_\sigma : \mathbb{R}^n \to \mathbb{R}^n$ is an image denoiser parameterized by $\sigma > 0$. Under additional technical assumptions, the solutions $\boldsymbol{x}^* \in \mathsf{zer}(\mathsf{G})$ can be associated with an explicit objective function of form (1). Specifically, when $\mathsf{D}_\sigma$ is locally homogeneous and has a symmetric Jacobian satisfying strong passivity (Romano et al., 2017; Reehorst & Schniter, 2019), $\mathsf{H}(\boldsymbol{x}) := \boldsymbol{x} - \mathsf{D}_\sigma(\boldsymbol{x})$ corresponds to the gradient of a convex regularizer

$$h(\boldsymbol{x}) = \frac{1}{2}\boldsymbol{x}^\mathsf{T}(\boldsymbol{x} - \mathsf{D}_\sigma(\boldsymbol{x})). \tag{3}$$

A simple strategy, known as GM-RED, for computing $\boldsymbol{x}^* \in \mathsf{zer}(\mathsf{G})$ is based on the first-order fixed-point iteration

$$\boldsymbol{x}^t = \boldsymbol{x}^{t-1} - \gamma\mathsf{G}(\boldsymbol{x}^{t-1}), \quad \text{with} \quad \mathsf{G} : \mathbb{R}^n \to \mathbb{R}^n, \tag{4}$$

where $\gamma > 0$ denotes the stepsize. In this paper, we extend this first-order RED algorithm to design ASYNC-RED. Since many denoisers do not satisfy the assumptions necessary for having an explicit objective (Reehorst & Schniter, 2019), our theoretical analysis considers a broader setting where $\mathsf{D}_\sigma$ does not necessarily correspond to any explicit regularizer. The benefit of our analysis is that it accommodates powerful deep denoisers (such as DnCNN (Zhang et al., 2017a)) that have been shown to achieve the state-of-the-art performance (Sun et al., 2019a; Wu et al., 2020; Cohen et al., 2020).

**Plug-and-play priors (PnP) and other related work.** There are other lines of works that combine the iterative methods with advanced denoisers. One closely-related framework is known as the *deep mean-shift priors* (Bigdeli et al., 2017). It develops an implicit regularizer whose gradient is specified by a denoising autoencoder. Another well-known framework is PnP, which generalizes proximal methods by replacing the proximal map with an image denoiser (Venkatakrishnan et al., 2013). Applications and theoretical analysis of PnP are widely studied in (Sreehari et al., 2016; Zhang et al., 2017b; Sun et al., 2019; Zhang et al., 2019; Ahmad et al., 2020; Wei et al., 2020) and (Chan et al., 2017; Meinhardt et al., 2017; Buzzard et al., 2018; Sun et al., 2019b; Tirer & Giryes, 2019; Teodoro et al., 2019; Ryu et al., 2019; Xu et al., 2020), respectively. In particular, Buzzard et al. (2018) proposed a parallel extension of PnP called *Consensus Equilibrium (CE)*, which enables synchronous parallel updates of $\boldsymbol{x}$. Note that while we developed ASYNC-RED as a variant of RED, our framework and analysis can be also potentially applied to PnP/CE. The plug-in strategy can be also applied to another family of algorithms known as *approximate message passing (AMP)* (Metzler et al., 2016a;b; Fletcher et al., 2018). The AMP-based algorithms are known to be nearly-optimal for random measurement matrices, but are generally unstable for general $\boldsymbol{A}$ (Rangan et al., 2014; 2015).

**Asynchronous parallel optimization.** There are two main lines of work in asynchronous parallel optimization, the one involving the asynchrony in coordinate updates (Iutzeler et al., 2013; Liu et al., 2015; Peng et al., 2016; Bianchi et al., 2015; Sun et al., 2017; Hannah & Yin, 2018; Hannah et al., 2019), and the other focusing on the study of various asynchronous stochastic gradient methods (Recht et al., 2011; Lian et al., 2015; Liu et al., 2018; Zhou et al., 2018; Lian et al., 2018).

Our work contributes to the area by developing a novel deep-regularized asynchronous parallel method with provable convergence guarantees.

## 3 ASYNCHRONOUS RED

ASYNC-RED allows efficient processing of data by simultaneously considering the asynchronous partial updates of solution $\boldsymbol{x}$ and the use of randomized subset of measurements $\boldsymbol{y}$. In this section, we introduce the algorithmic details of our method. We start with the basic batch formulation of ASYNC-RED (ASYNC-RED-BG) followed by its minibatch variant (ASYNC-RED-SG).

### 3.1 ASYNC-RED USING BATCH GRADIENT

When the gradient uses all the measurements $\boldsymbol{y} \in \mathbb{R}^m$, ASYNC-RED-BG is the *asynchronous* extension of the recent block-coordinate RED (BC-RED) algorithm (Sun et al., 2019a). Consider the decomposition of the variable space $\mathbb{R}^n$ into $b \geq 1$ blocks

$$\boldsymbol{x} = (\boldsymbol{x}_1, \cdots, \boldsymbol{x}_b) \in \mathbb{R}^{n_1} \times \cdots \times \mathbb{R}^{n_b} = \mathbb{R}^n \quad \text{with} \quad n = n_1 + n_2 + \cdots + n_b,$$

For each $i \in \{1, \ldots, b\}$, we introduce the operator $\mathsf{U}_i : \mathbb{R}^{n_i} \to \mathbb{R}^n$ that injects a vector in $\mathbb{R}^{n_i}$ into $\mathbb{R}^n$ and its transpose $\mathsf{U}_i^\mathsf{T}$ that extracts the $i$th block from a vector in $\mathbb{R}^n$. This directly implies that

$$\mathsf{I} = \mathsf{U}_1 \mathsf{U}_1^\mathsf{T} + \cdots + \mathsf{U}_b \mathsf{U}_b^\mathsf{T} \quad \text{and} \quad \|\boldsymbol{x}\|_2^2 = \|\boldsymbol{x}_1\|_2^2 + \cdots + \|\boldsymbol{x}_b\|_2^2 \quad \text{with} \quad \boldsymbol{x}_i = \mathsf{U}_i^\mathsf{T} \boldsymbol{x}. \quad (5)$$

In analogy to the RED operator $\mathsf{G}$ in (2), we define the block-coordinate operator $\mathsf{G}_i$ as

$$\mathsf{G}_i(\boldsymbol{x}) \coloneqq \mathsf{U}_i \mathsf{U}_i^\mathsf{T} \mathsf{G}(\boldsymbol{x}), \quad \text{with} \quad \boldsymbol{x} \in \mathbb{R}^n \quad \text{and} \quad \mathsf{G}_i : \mathbb{R}^n \to \mathbb{R}^n. \quad (6)$$

Due to the asynchrony in the block updates, the iterate might be updated several times by different cores during a single update cycle of a core, which means that the evaluation of $\boldsymbol{x}^{k+1}$ relies on a *stale* iterate $\widetilde{\boldsymbol{x}}^k$

$$\boldsymbol{x}^{k+1} \leftarrow \boldsymbol{x}^k - \gamma \mathsf{G}_{i_k}(\widetilde{\boldsymbol{x}}^k), \quad \text{with} \quad \widetilde{\boldsymbol{x}}^k = \boldsymbol{x}^k + \sum_{s=k-\Delta_k}^{k-1} (\boldsymbol{x}^s - \boldsymbol{x}^{s+1}), \quad \Delta_k \leq \lambda. \quad (7)$$

Here, we assume that the stale iterate $\widetilde{\boldsymbol{x}}^k$ exits as a state of $\boldsymbol{x}$ in the shared memory, and the delay between them is bounded by a finite number $\lambda \in \mathbb{Z}_+$. These two assumptions are often referred to as the *consistent read* (Recht et al., 2011) and the *bounded delay* (Liu & Wright, 2015) in the traditional asynchronous block coordinate optimization. Although we implement the consistent read in ASYNC-RED, the algorithm never imposes a global lock on $\boldsymbol{x}^k$. We refer to Supplement A for the related discussion.

---

**Algorithm 1** ASYNC-RED-BG

---

1: **input:** $\boldsymbol{x}^0 \in \mathbb{R}^n, \gamma > 0, \tau > 0$.
2: **setup:** A multicore system with one shared memory storing $\boldsymbol{x}$ and global iteration $k$.
3: **for global** $k = 1, 2, 3, \ldots$ **do**
4:     $\widetilde{\boldsymbol{x}}^k \leftarrow \mathsf{read}(\boldsymbol{x})$
5:     $\mathsf{G}_{i_k}(\widetilde{\boldsymbol{x}}^k) \leftarrow \mathsf{U}_{i_k}\mathsf{U}_{i_k}^\mathsf{T}\mathsf{G}(\widetilde{\boldsymbol{x}}^k)$     with random $i_k \in \{1, \ldots, b\}$          ▷ Block Operation
6:     $\boldsymbol{x}^k \leftarrow \mathsf{read}(\boldsymbol{x})$
7:     $\boldsymbol{x}^{k+1} \leftarrow \boldsymbol{x}^k - \gamma\mathsf{G}_{i_k}(\widetilde{\boldsymbol{x}}^k)$
8:     update $\boldsymbol{x}$ in the shared memory using $\boldsymbol{x}^{k+1}$
9: **end for**

---

The first variant, ASYNC-RED-BG, is summarized in Algorithm 1, where $\mathsf{read}(\cdot)$ reads a block from the shared memory to the local memory. When the algorithm is run on a single core system without parallelization (that is to say $\widetilde{\boldsymbol{x}}^k = \boldsymbol{x}^k$), it reduces to the normal BC-RED algorithm. Hence, our analysis is also applicable to BC-RED.

We specifically consider the *random* block selection strategy in ASYNC-RED-BG, namely that every block index $i_k$ is selected as an i.i.d random variable uniformly distributed over $\{1, \ldots, b\}$. Such a strategy is commonly adopted for simplifying the convergence analysis. Nevertheless, our method and analysis can be generalized to the scenario where $i_k$ follows some arbitrary probability $P(i_k = i) = p_i$ specified by the user.

Compared with serial RED algorithms, ASYNC-RED-BG enjoys considerable scalability by dividing the computation of the full operator $\mathsf{G}$ into $b$ parallel evaluation of $\mathsf{G}_i$ distributed across all cores. Thus, without any modification to the algorithmic design, one can easily improve the performance of the algorithm by simply integrating more cores into the system. In Section 5, we experimentally demonstrate the significant speed-up and scale-up in solving the context of image recovery.

## 3.2 ASYNC-RED USING STOCHASTIC GRADIENT

The scale of measurements is another important factor influencing the computational complexity in the large-scale inference tasks. ASYNC-RED-SG improves the applicability of ASYNC-RED to these cases by further considering the decomposition of the measurement space $\mathbb{R}^m$ into $\ell \geq 1$ blocks

$$\boldsymbol{y} = (\boldsymbol{y}_1, \cdots, \boldsymbol{y}_\ell) \in \mathbb{R}^{m_1} \times \cdots \times \mathbb{R}^{m_\ell} = \mathbb{R}^m \quad \text{with} \quad m = m_1 + m_2 + \cdots + m_\ell.$$

Hence, ASYNC-RED-SG considers the following data-fidelity $g$ and its gradient $\nabla g$

$$g(\boldsymbol{x}) = \frac{1}{\ell}\sum_{j=1}^{\ell} g_j(\boldsymbol{x}) \quad \Rightarrow \quad \nabla g(\boldsymbol{x}) = \frac{1}{\ell}\sum_{j=1}^{\ell} \nabla g_j(\boldsymbol{x}), \tag{8}$$

where each $g_j$ is evaluated on the subset $\boldsymbol{y}_j \in \mathbb{R}^{m_j}$ of the full $\boldsymbol{y}$. From (8), we know that the computation of $\nabla g(\boldsymbol{x})$ is proportional to the total number $\ell$. To reduce the per-iteration cost, we follow the idea of stochastic optimization to approximate the batch gradient by using the stochastic gradient that relies on a minibatch of $w \ll \ell$ measurements

$$\widehat{\nabla} g(\boldsymbol{x}) = \frac{1}{w}\sum_{s=1}^{w} \nabla g_{j_s}(\boldsymbol{x}), \tag{9}$$

where $j_s$ is picked from the set $\{1, \ldots, \ell\}$ as i.i.d uniform random variable. Based on the minibatch gradient, we define the block stochastic operator $\widehat{\mathsf{G}}_i : \mathbb{R}^n \to \mathbb{R}^n$ as

$$\widehat{\mathsf{G}}_i := \mathsf{U}_i\mathsf{U}_i^\mathsf{T}\widehat{\mathsf{G}}(\boldsymbol{x}), \quad \text{with} \quad \widehat{\mathsf{G}} := \widehat{\nabla} g(\boldsymbol{x}) + \tau(\boldsymbol{x} - \mathsf{D}_\sigma(\boldsymbol{x})), \quad \widehat{\mathsf{G}} : \mathbb{R}^n \to \mathbb{R}^n. \tag{10}$$

Note that the computation of $\widehat{\mathsf{G}}_i$ is now dependent on the minibatch size $w$ that is adjustable to cope with the computation resources at hand. ASYNC-RED-SG is summarized in Algorithm 2.

The operation $\mathsf{minibatchG}(\cdot)$ computes the estimate of $\mathsf{G}$ based on $w$ randomly selected measurements. We clarify the difference between ASYNC-RED-SG and ASYNC-RED-BG via a specific example.

---

**Algorithm 2** ASYNC-RED-SG

---

1: **input:** $\boldsymbol{x}^0 \in \mathbb{R}^n, \gamma > 0, \tau > 0$.
2: **setup:** A multicore system with one shared memory storing $\boldsymbol{x}$ and global iteration $k$.
3: **for global** $k = 1, 2, 3, \dots$ **do**
4:      $\widetilde{\boldsymbol{x}}^k \leftarrow \mathsf{read}(\boldsymbol{x})$
5:      $\widehat{\mathsf{G}}(\widetilde{\boldsymbol{x}}^k) \leftarrow \mathsf{minibatchG}(\widetilde{\boldsymbol{x}}^k, w)$    with random $j_w \in \{1, \dots, \ell\}$           ▷ Minibatch Gradient
6:      $\widehat{\mathsf{G}}_{i_k}(\widetilde{\boldsymbol{x}}^k) \leftarrow \mathsf{U}_{i_k} \mathsf{U}_{i_k}^\mathsf{T} \widehat{\mathsf{G}}(\widetilde{\boldsymbol{x}}^k)$    with random $i_k \in \{1, \dots, b\}$           ▷ Block Operation
7:      $\boldsymbol{x}^k \leftarrow \mathsf{read}(\boldsymbol{x})$
8:      $\boldsymbol{x}^{k+1} \leftarrow \boldsymbol{x}^k - \gamma \widehat{\mathsf{G}}_{i_k}(\widetilde{\boldsymbol{x}}^k)$
9:      update $\boldsymbol{x}$ in the shared memory using $\boldsymbol{x}^{k+1}$
10: **end for**

---

Consider the least-squares $g$ with a block-friendly operator $\boldsymbol{A}$ and a block-efficient denoiser $\mathsf{D}_\sigma$. We can write the update of ASYNC-RED-BG regarding a single iteration as

$$\mathsf{G}_i(\widetilde{\boldsymbol{x}}) = \boldsymbol{A}_i^\mathsf{T}(\boldsymbol{A}_i \widetilde{\boldsymbol{x}} - \boldsymbol{y}_i) + \tau(\widetilde{\boldsymbol{x}}_i - \mathsf{D}(\widetilde{\boldsymbol{x}}_i)), \tag{11}$$

where $\widetilde{\boldsymbol{x}}$ is the delayed iterate for $\boldsymbol{x}$, and $\boldsymbol{A}_i \in \mathbb{R}^{m \times n_i}$ is a submatrix of $\boldsymbol{A}$ consisting of columns corresponding to the $i$th blocks. Although the per-iteration complexity is reduced by roughly $b = n/n_i$ times by working with $\boldsymbol{A}_i$ instead of $\boldsymbol{A}$, ASYNC-RED-BG still needs to work with all the measurements $\boldsymbol{y}_i$ related to the $i$th block at every iteration. Consider the corresponding update of ASYNC-RED-SG with one measurement used at a time

$$\widehat{\mathsf{G}}_i(\widetilde{\boldsymbol{x}}) = \boldsymbol{A}_{ji}^\mathsf{T}(\boldsymbol{A}_{ji} \widetilde{\boldsymbol{x}} - \boldsymbol{y}_{ji}) + \tau(\widetilde{\boldsymbol{x}}_i - \mathsf{D}(\widetilde{\boldsymbol{x}}_i)), \tag{12}$$

where $\boldsymbol{y}_{ji}$ denotes the $j$th measurement of $\boldsymbol{x}_i$, and $\boldsymbol{A}_{ji} \in \mathbb{R}^{m_j \times n_i}$ is the submatrix crossed by the rows and columns corresponding to the $j$th measurement and the $i$th blocks. This indicates that the reduction of the per-iteration complexity from ASYNC-RED-BG to ASYNC-RED-SG can be up to $\ell = m/m_j$ times. In the practice, it is common to use $w > 1$ measurements at a time to optimize the total runtime. Note that if $\mathsf{U} = \mathsf{U}^\mathsf{T} = \mathsf{I}$, ASYNC-RED-SG becomes the asynchronous stochastic RED algorithm. In the next section, we will present a complete analysis of ASYNC-RED and theoretically discuss its connection to the related algorithms.

## 4 CONVERGENCE ANALYSIS OF ASYNC-RED

The proposed analysis is based on the following explicit assumptions. Note that these assumptions serve as sufficient conditions for the convergence.

**Assumption 1.** *We assume bounded maximal delay $\lambda < \infty$. Hence, during any update cycle of an agent, the estimate $\boldsymbol{x}$ in the shared memory is updated at most $\lambda \in \mathbb{Z}_+$ times by other cores.*

The value of $\lambda$ is often dependent on the number of cores involved in the computation (Wright, 2015). If every core takes a similar amount of time to compute its update, $\lambda$ is expected to be a multiple of the number of cores. Related work has investigated the convergence with unbounded maximal delays in the context of traditional optimization (Hannah & Yin, 2018; Peng et al., 2019; Zhou et al., 2018).

**Assumption 2.** *The operator $\mathsf{G}$ is such that $\mathsf{zer}(\mathsf{G}) \neq \varnothing$, and the distance of the initial $\boldsymbol{x}^0 \in \mathbb{R}^n$ to any element in $\mathsf{zer}(\mathsf{G})$ is bounded, that is $\|\boldsymbol{x}^0 - \boldsymbol{x}^*\| \leq R_0$ for all $\boldsymbol{x}^* \in \mathsf{zer}(\mathsf{G})$ with $R_0 < \infty$.*

This assumption ensures the existence of a solution for the RED problem and is related to the existence of minimizers in traditional coordinate minimization (Nesterov, 2012; Beck & Tetruashvili, 2013)

**Assumption 3.** *(a) Every component function $g_i$ is convex differentiable and has a Lipschitz continuous gradient of constant $L_i > 0$. (b) At every update, the stochastic gradient is unbiased estimator of $\nabla g$ that has a bounded variance:*

$$\mathbb{E}\left[\widehat{\nabla} g(\boldsymbol{x})\right] = g(\boldsymbol{x}), \quad \mathbb{E}\left[\|\widehat{\nabla} g(\boldsymbol{x}) - \nabla g(\boldsymbol{x})\|^2\right] \leq \frac{\nu^2}{w}, \quad \boldsymbol{x} \in \mathbb{R}^n, \quad \nu > 0.$$

The first part of the assumption implies that $g$ is also convex and has Lipschitz continuous gradient with constant $L = \mathsf{max}\{L_1, \dots, L_\ell\}$. The second part is a standard assumption on the unbiasedness

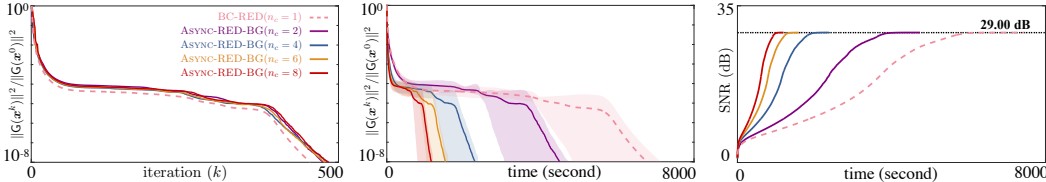

Figure 2: Convergence of ASYNC-RED-BG for different numbers of accessible cores $n_c \in \{2, 4, 6, 8\}$. The left figure plots the average normalized distance to zer(G) against the iteration number; the middle and right figures plot these values, as well as SNR, plotted against the actual runtime in seconds. The shaded areas represent the range of values attained over the test images.

| Method | SNR | time | speed-up |
|---|---|---|---|
| SYNC-RED(8-core) | 29.00 dB | 38.9 min | 2.8× |
| ASYNC-RED-BG(8-core) | **29.01 dB** | 17.9 min | 6.1× |
| **Async-RED-SG(8-core)** | 28.08 dB | **13.0 min** | **8.4×** |

and variance of the stochastic gradient (Lian et al., 2015; Ghadimi & Lan, 2016). Our final assumption is related to the deep denoiser used in ASYNC-RED.

**Assumption 4.** *The denoiser $D_\sigma$ is a nonexpansive operator $\|D_\sigma(\boldsymbol{x}) - D_\sigma(\boldsymbol{y})\| \leq \|\boldsymbol{x} - \boldsymbol{y}\|$.*

Compared with the conditions stated in Section 2 (namely, that it is locally homogeneous with a symmetric Jacobian), our requirement on the denoiser is milder. One can train a nonexpansive $D_\sigma$ by constraining the Lipschitz constant of $D_\sigma$ via the spectral normalization, which is an active area of research in deep learning (Miyato et al., 2018; Sedghi et al., 2019; Anil et al., 2019; Terris et al., 2020).

We can now state the theorems on ASYNC-RED.

**Theorem 1.** *Let Assumptions 1-4 hold true. Run ASYNC-RED-BG for $t > 0$ iterations with uniform i.i.d block selection using a fixed step-size $\gamma \in (0, 1/((1 + 2\lambda)(L + 2\tau)))$. Then, the iterates of the algorithm satisfy*

$$\min_{0 \leq k \leq t-1} \mathbb{E}\left[\|G(\boldsymbol{x}^k)\|^2\right] \leq \left[\frac{D}{b} + 2\right] \frac{(L + 2\tau)b}{\gamma t} R_0^2. \tag{13}$$

*where $D = 2\lambda^2/(1 + \lambda)^2$ is a constant.*

Theorem 1 establishes the convergence of ASYNC-RED-BG to the fixed-point set zer(G) at the rate of $O(1/t)$. Our result is consistent with the existing results in the literature. In particular, when the algorithm adopts serial block updates, that is $\lambda = 0$ and $\widetilde{\boldsymbol{x}}^k = \boldsymbol{x}^k$, the recovered convergence is nearly the same as BC-RED (Sun et al., 2019a) scaled by some constant. On the other hand, our convergence rate $O(1/t)$ is also consistent with the rate proved for the asynchronous block coordinate descent in nonconvex optimization (Sun et al., 2017).

**Theorem 2.** *Let Assumptions 1-4 hold true. Run ASYNC-RED-SG for $t > 0$ iterations with uniform i.i.d selections of blocks and measurements using a fixed step-size $\gamma \in (0, 1/((1 + 2\lambda)(L + 2\tau)))$. Then, the iterates of the algorithm satisfy*

$$\min_{0 \leq k \leq t-1} \mathbb{E}\left[\|G(\boldsymbol{x}^k)\|^2\right] \leq \left[\frac{D}{b} + 2\right] \frac{(L + 2\tau)b}{\gamma t} R_0^2 + \left[\frac{2D}{b} + 2\right] \frac{\gamma}{w} C \tag{14}$$

*where $C = (L + 2\tau)(1 + \lambda)\nu^2$ and $D = 2\lambda^2/(1 + \lambda)^2$ are constants.*

Theorem 2 states that ASYNC-RED-SG approximates the solution obtained by ASYNC-RED-BG up to a finite error that decreases for larger values of the minibatch size $w$. This relationship is consistent with the recent theoretical results on the online PnP and RED algorithms (Sun et al., 2019b; Wu et al., 2020). In practice, the selection of $w$ must balance the actual memory capacity of the system and the desired runtime for obtaining a reasonable solution. Our numerical evaluation in Section 5 demonstrates the excellent approximation of ASYNC-RED-SG to the batch-gradient solution by using a small subset of data.

By carefully choosing the stepsize $\gamma$, we can state the following remark on Theorem 2.
**Remark 1.** Set the stepsize to be $\gamma = 1/\sqrt{wt}$. If the maximal delay satisfies $\lambda \leq (1/2)[\sqrt{wt}/(L + 2\tau) - 1]$, then after $t > 0$ iterations we have

$$\min_{0 \leq k \leq t-1} \mathbb{E}\left[\|G(\boldsymbol{x}^k)\|^2\right] \leq \left[\frac{D}{b} + 2\right] \frac{(L + 2\tau)b}{\sqrt{wt}} R_0^2 + \left[\frac{2D}{b} + 2\right] \frac{C}{\sqrt{wt}}. \tag{15}$$

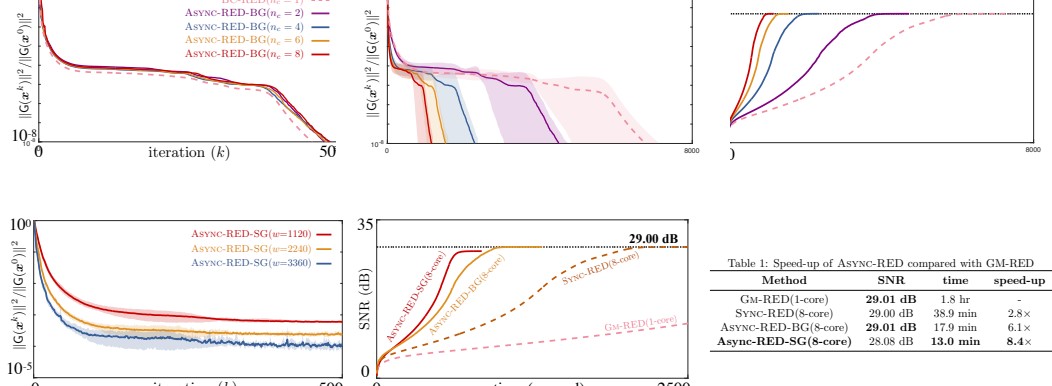

Table 1: Speed-up of ASYNC-RED compared with GM-RED

| Method | SNR | time | speed-up |
|---|---|---|---|
| GM-RED(1-core) | **29.01 dB** | 1.8 hr | - |
| SYNC-RED(8-core) | 29.00 dB | 38.9 min | 2.8× |
| ASYNC-RED-BG(8-core) | **29.01 dB** | 17.9 min | 6.1× |
| **ASYNC-RED-SG(8-core)** | 28.08 dB | **13.0 min** | **8.4×** |

Figure 3: *Left:* Evolution of the convergence accuracy of ASYNC-RED-SG as the minibatch size $w$ increases. The average distance is plotted against the number of iterations with the shaded areas representing the range of values attained over the test images. *Middle & Right:* Comparison of convergence speed between ASYNC-RED-BG/SG and other baselines. The right table summarizes the total runtime and the speed-up compared with GM-RED for all algorithms.

This establishes the fixed-point convergence to the set zer(G) at the rate of $O(1/\sqrt{wt})$ under specific conditions. If we treat entire $x$ as a block, namely that $\mathsf{U} = \mathsf{U}^\mathsf{T} = \mathsf{I}$ and $b = 1$, ASYNC-RED-SG then becomes the asynchronous stochastic RED algorithm. Hence, the proposed remark immediately holds true for the later. Note that our convergence rate $O(1/\sqrt{wt})$ is consistent with the rate proved for the serial (Nemirovski et al., 2009) and parallel (Dekel et al., 2012; Lian et al., 2015) stochastic gradient methods.

All the proofs are presented in the supplement. We note that the analysis above does not assume the existence of an explicit regularizer associated with the operator $\mathsf{D}_\sigma$. Moreover, it does not require $\mathsf{D}_\sigma$ to be a Gaussian denoiser. Our analysis is hence applicable to all nonexpansive operators, such as the traditional proximal operators or the more recent artifact-removal operators (Zhang et al., 2019).

## 5 NUMERICAL VALIDATION

We now present a numerical validation of ASYNC-RED. Our goals are first to validate the proposed theorems in Section 4 and then to demonstrate the effectiveness and the efficiency of our algorithm on the large-scale problem. We consider two image recovery tasks that have the form $y = Ax + e$, where the measurement matrix $A$ corresponds to either the random matrix in *compressive sensing (CS)* or the Radon transform in *computed tomography (CT)*, and the noise $e$ is assumed to be additive white Gaussian (AWGN). In particular, the random matrix is implemented with the block-diagonal structure $A = \mathsf{diag}([A_i, ..., A_b])$ for fast validation, while the Radon transform is used as its full matrix form to demonstrate the effectiveness of ASYNC-RED for overcoming the computation bottleneck. Our deep neural net prior adapts the DnCNN architecture (Zhang et al., 2017a). We used the signal-to-noise ratio (dB) to quantify the quality of the reconstructed images. For each experiments, we selected the denoiser that achieves the best SNR performance from the ones corresponding to five noise levels $\sigma \in \{5, 10, 15, 20, 25\}$. The value of $\sigma$ is fixed across all iterations of the algorithm. Supplement D provides additional technical details.

### 5.1 CONVERGENCE BEHAVIOR

We validate our theorems on the CS task with 6 test images selected from the *Set 12* dataset (Zhang et al., 2017a). Each test image is rescaled to the size of $240 \times 240$ pixels (see Fig. 6 in the supplement for the visualization). The block-diagonal matrix $A$ is set to consist of 9 submatrices, corresponding to a $3 \times 3$ grid of blocks with the size of $80 \times 80$ pixels in every image. The elements in $A$ are i.i.d zero-mean Gaussian random variables of variance of $1/m$, and the compression ratio is set to be $m/n = 0.7$, which indicates that the total number of measurements is 4480 for each block. We obtain the measurements by multiplying $A$ with each vectorized image and adding additional noise corresponding to the input SNR of 30 dB. Finally, we use the normalized distance $\|\mathsf{G}(x^k)\|_2^2/\|\mathsf{G}(x^0)\|_2^2$ to quantify the fixed-point convergence, with $b$ block updates grouped as one iteration. The distance is expected to approach zero as the algorithm converges to a fixed point. The average performance of all methods is obtained by running a single trial for each image.

Theorem 1 establishes the convergence of ASYNC-RED-BG to the fixed point set zer(G). This is illustrated in Fig. 2 for four different numbers of accessible cores $n_c \in \{2, 4, 6, 8\}$. In the left figure, the average normalized distance is plotted against the iteration number, while the middle and

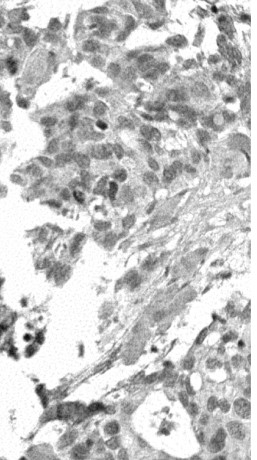

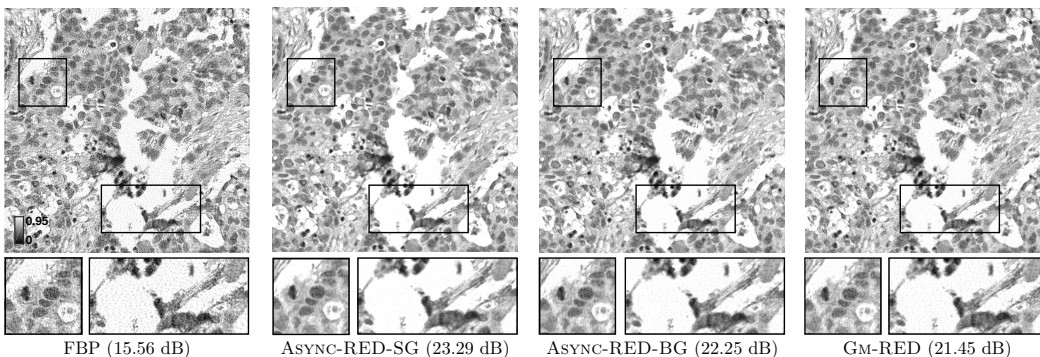

| FBP (15.56 dB) | ASYNC-RED-SG (23.29 dB) | ASYNC-RED-BG (22.25 dB) | GM-RED (21.45 dB) |

Figure 4: CT reconstruction with a time budget of 1 hour by ASYNC-RED-BG/SG and GM-RED. The colormap is adjusted for the best visual quality.

right figures plot the corresponding distance and SNR values against the actual runtime in seconds. The shaded areas representing the range of values attained across all test images. We also plot the results of serial BC-RED using the dashed line as reference. ASYNC-RED-BG is implemented to be run asynchronously on multiple cores, while BC-RED can only use one core to perform the computation. The left figure highlights the fixed-point convergence of ASYNC-RED-BG in iteration for different $n_c$, with all variants agreeing with the serial BC-RED. Since ASYNC-RED-BG uses more cores, the middle and right figures demonstrate the significantly faster in-time convergence of ASYNC-RED-BG than BC-RED to the same SNR value. Specifically, BC-RED takes 1.8 hours to achieve 29.00 dB, while ASYNC-RED-BG ($n_c = 8$) takes only 17.9 minutes to obtain the same value, corresponding to a $6\times$ improvement in computation time.

Theorem 2 establishes the convergence of ASYNC-RED-SG to zer(G) up to some error term, which is inversely proportional to the minibatch size $w$. This is illustrated in Fig. 3 (left) for three different minibatch sizes $w \in \{1120, 2240, 3360\}$. As before, we plotted the average distance against the iteration number with the shading area representing the variance. Note that the log-scale of y-axis highlights the change for smaller values. Fig. 3 demonstrates the improved convergence of ASYNC-RED-SG to zer(G) for larger $w$, which is consistent with our theoretical analysis. Fig. 3 (middle) compares the convergence speed between ASYNC-RED-BG/SG, gradient-method RED (GM-RED), and synchronous parallel RED (SYNC-RED). For ASYNC-RED-SG, we use $w = 1120$. In particular, ASYNC-RED-SG takes fewer total runtime (from 17.9 min to 13.0 min) to obtain the similar result (29.01 dB and 28.03 dB) and achieves $8.4\times$ speedup compared with GM-RED. The table in Fig. 3 summarizes the detailed results.

## 5.2 EFFECTIVENESS FOR COMPUTATIONAL IMAGING

We additionally demonstrate the effectiveness of our algorithm by reconstructing a $800 \times 800$ CT image from its 180 projections. For block parallel updates, the image is decomposed into 16 blocks, each having the size of $200 \times 200$ pixels. The Radon matrix used in the experiment corresponds to 180 angles with 1131 detectors, and the noise level is set to 70 dB. We refer to Supplement D.2 for additional technical details. Fig. 4 shows the visual illustration of the reconstructed images by ASYNC-RED-BG/SG and GM-RED. Each algorithm starts from the filtered back-projection (FBP) of the measurements and runs for 1 hour. Here, ASYNC-RED-SG randomly uses one-third of the total measurements at every iteration. Given the same amount of time, ASYNC-RED-BG/SG successfully mitigates the noise-artifacts, while the result of GM-RED is still noisy. In particular, the per-iteration time cost of ASYNC-RED-BG/SG and GM-RED is 5.23, 3.21, and 19.19 seconds, respectively. This experiment clearly illustrates the fast processing speed of the asynchronous procedure.

## 6 CONCLUSION

Asynchronous parallel methods have gained increasing importance in optimization for solving large-scale imaging inverse problems. We have introduced ASYNC-RED as an extension of the recent RED framework and theoretically analyze its convergence in batch and stochastic settings. We have

validated its convergence guarantees and demonstrated its effectiveness in CT image reconstruction. We note that this work is complementary to traditional acceleration strategies, such as Nesterov acceleration and variance-reduction, commonly used in optimization. Future work will investigate ASYNC-RED with Nesterov acceleration (as was done in (Hannah et al., 2019) for traditional asynchronous block-coordinate algorithms) and variance-reduction (as was done in (Johnson & Zhang, 2013) for traditional stochastic gradient method) to better understand the tradeoffs between acceleration and scalability in multicore systems. We will additionally investigate theoretical limits of ASYNC-RED in the unbounded maximal delay setting.

## ACKNOWLEDGEMENT

Research presented in this article was supported by NSF award CCF-1813910 and the Laboratory Directed Research and Development program of Los Alamos National Laboratory under project number 20200061DR.

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
