# OpenReview forum: "Async-RED: A Provably Convergent Asynchronous Block Parallel Stochastic Method using Deep Denoising Priors"
_ICLR.cc/2021/Conference — ICLR 2021 Spotlight_

### Official Review · AnonReviewer4 · 2020-10-21
**A good paper on accelerating distributed computations by asynchronous and stochastic activation, with applications to regularized imaging problems**

**Rating:** 7
**Confidence:** 3

**Review:**

The main contribution is a combination of asynchronous processing and stochastic activation of blocks in a distributed computing environment. The framework is general, since the regularization/denoising operator is any nonexpansive operator; but focusing on the PnP/RED framework is a very relevant choice.
To my knowledge (I am not an expert of this particular area of optimization), the contribution is new and solid.
I have the following questions/remarks:
* After Assumption 4: when mentioning the literature, I think you should include the paper "BUILDING FIRMLY NONEXPANSIVE CONVOLUTIONAL NEURAL NETWORKS" by Terris et al. and shortly discuss the relationship. In particular, nonexpansiveness is sufficient in your setting, whereas it is usually not sufficient in optimization, with averagedness/firm nonexpansiveness assumed. Why is it so?
* About asynchronous optimization, Franck Iutzeler has his PhD thesis and several papers on the topic. You might have a look and cite some of them, which are relevant in your setting.
* Theorem 2 shows that the "convergence" is not variance-reduced. I would appreciate a discussion on whether this is unavoidable or if this is an open question for future work, if there is relevant literature on this matter, and why it is difficult to derive a variance-reduced approach with similar features.
* the DnCNN architecture is used in the experiments: does it satisfy Assumption 4? More generally, are all assumptions met so that the theorems apply in the experiments? You should discuss the match between your theoretical results and the conditions of the practical experiments more closely.

---

> ### Author Response · Authors · 2020-11-18
> **Response to Reviewer4**
>
> **1. After Assumption 4: when mentioning the literature, I think you should include the paper “BUILDING FIRMLY NONEXPANSIVE CONVOLUTIONAL NEURAL NETWORKS” by Terris et al. and shortly discuss the relationship.**
> * We cite the paper in the updated manuscript.
>
> **2. In particular, nonexpansiveness is sufficient in your setting, whereas it is usually not sufficient in
> optimization, with averagedness/firm nonexpansiveness assumed. Why is it so?**
> * This is the direct consequence of the way we formulate our operators. If denoiser $D_\sigma$ is nonexpansive, then according to Lemma 1 in Supplement B.1, we know that the composite operator $G := I − D_\sigma$ is $1/(L + 2\tau)$-cocoercive. Consider the inequality in Eq. 20 in Supplement B.1,  where the cross term $\dagger$ can be bounded by using the cocoercivity of $G$. The expression of the final bound can be found in Eq. 26 in Supplement B.1. Once we obtain the bound for the expectation, we then can take the total expectation and apply the telescopic-sum over 1,...,t iterations to establish the convergence (Eq. 27-Eq. 39). Similar formulations can be found in the analysis of BC-RED.
>
> **3. About asynchronous optimization, Franck Iutzeler has his PhD thesis and several papers on the topic. You might have a look and cite some of them, which are relevant in your setting.**
> * We have reviewed several papers by Franck Lutzeler and cite them in the updated manuscript.
>
> **4. Theorem 2 shows that the “convergence” is not variance-reduced. I would appreciate a discussion on whether this is unavoidable or if this is an open question for future work, if there is relevant literature on this matter, and why it is difficult to derive a variance-reduced approach with similar features.**
> * We have added the discussion on p.6 of the revised manuscript.
>
> **5. The DnCNN architecture is used in the experiments: does it satisfy Assumption 4? More generally, are all assumptions met so that the theorems apply in the experiments? You should discuss the match between your theoretical results and the conditions of the practical experiments more closely.**
> * In Supplement D.1, we provide more technical details on the DnCNN architecture and its training strategy. We adopt the spectral normalization technique from [1] to control the Lipschitz constant (LC) of our DnCNN prior. In the training, we constrain the residual network $R_\sigma$ (see Fig. 5 in the Supplement) such that its LC is smaller than 2. Since the non-expansiveness of $D_\sigma$ implies that Rσ has LC ≤ 2, this provides a necessary condition for $D_\sigma$ to satisfy Assumption 4. The effectiveness of this strategy was also discussed in BC-RED paper, where it was shown that without the constraint on LC of DnCNN the algorithm can diverge. Finally, note that all the other assumptions in our analysis are also satisfied in our experiments, which has been explicitly stated in the revision.
>
> **Reference**
>
> [1] H. Sedghi, V. Gupta, and P. M. Long. The singular values of convolutional layers. In International Conference on Learning Representations, 2019.

---

### Official Review · AnonReviewer2 · 2020-10-24
**First asynchronous regularization-by-denoising algorithms for imaging with deep denoising priors**

**Rating:** 7
**Confidence:** 5

**Review:**

This paper proposed for the first time the asynchronous variants of deterministic and stochastic regularization-by-denoising (RED) algorithms which have become popular recently in image recovery and reconstruction applications since they leverage the power of  pretrained deep denoising neural networks into the traditional model-based schemes, and often achieve state-of-the-art recovery results. These new variants are aimed to fully utilize the multi-cores structure of computational devices and to maximize the practicality of PnP/RED methods in large-scale inverse problems. The authors provide gradient-norm convergence analysis for the proposed algorithms under standard assumptions, along with detailed numerical studies demonstrating practical advantageous of proposed methods.

Overall the paper is well-written and has the potential to be a nice contribution to the community. The reviewer has several concerns about the current version which hopefully the authors would address and clarify:

(1) The proposed Async-RED methods apply the denoiser on blocks of the image, while the original RED applies the denoiser on the whole image. Intuitively, such a scheme may be suboptimal and somewhat compromise the recovery performance, since state-of-the-art DNN denoisers utilize non-local similarity across the image. Although the experiments in this work do not demonstrate such deficit, the reviewer suspects that, for some images, the Async-RED may not do well on the pixels near the edges of the blocks. The reviewer is a bit dubious about whether such type of block decomposition of denoiser in this line of work can be universally reliable.

(2) The comparative results shown in Fig 3 regarding compressed-sensing reconstruction appear to be somewhat unconvincing:

Firstly, the authors use a suboptimal GM-RED/Sync-RED for baseline, which is gradient descent RED without Nesterov acceleration, while it is well-known that the accelerated gradient descent  RED (AGD-RED) converges at a much better rate and the reviewer believes that this is a more sensible baseline. Maybe an important advantage of original RED and synchronous RED is that they are easier to be accelerated by momentum tricks?

Secondly, Async-RED-SG seems to yield very limited acceleration in terms of convergence rate over the deterministic counterpart Async-RED-BG in CS example -- what is the minibatch size used in this experiment? Would reducing the minibatch size yield a faster convergence rate? Meanwhile the recovery result of Async-RED-SG is 1dB worse than the other methods, which seems a bit problematic (would a shrinking step-size help to fix this issue?).

It is highly suggested that the authors should also plot the convergence curves for the CT experiments as in the Fig.3.

******************** after rebuttal*********************

The authors have provided a responsive feedback and addressed the comments to satisfactory, therefore the reviewer votes for acceptance.

---

> ### Author Response · Authors · 2020-11-18
> **Response to Reviewer2**
>
> **1. The proposed Async-RED methods apply the denoiser on blocks of the image, while the original RED applies the denoiser on the whole image. Intuitively, such a scheme may be suboptimal and somewhat compromise the recovery performance, since state-of-the-art DNN denoisers utilize non-local similarity across the image. Although the experiments in this work do not demonstrate such deficit, the reviewer suspects that, for some images, the Async-RED may not do well on the pixels near the edges of the blocks. The reviewer is a bit dubious about whether such type of block decomposition of denoiser in this line of work can be universally reliable.**
> * This is an excellent remark worth a discussion in the revised manuscript. Let us start by mentioning that the block-wise nature of Async-RED is by no means universally applicable; however, the algorithm and its analysis are compatible with a variety of block decomposition strategies (e.g. by rows or columns of the image) that might be useful for different applications. This feature is consistent with the traditional block coordinate methods [1]. We would also like to point out that a simple strategy for mitigating artifacts at the block edges is to include additional neighboring pixels at the input of the DNN (but use the exact block size at the output). Table 3 (see additional_results.pdf in the supplement) reports our results of experimenting with this idea of input padding. The results indicate that by including a small number of pixels around each block at the input of DNN, one can match the performance of using the full image at the input of DNN. This is included in the supplement in the revision.
>
> **2. The comparative results shown in Fig 3 regarding compressed-sensing reconstruction appear to be somewhat unconvincing: Firstly, the authors use a suboptimal Gm-RED/Sync-RED for baseline, which is gradient descent RED without Nesterov acceleration, while it is well-known that the accelerated gradient descent RED (AGD-RED) converges at a much better rate and the reviewer believes that this is a more sensible baseline. Maybe an important advantage of original RED and synchronous RED is that they are easier to be accelerated by momentum tricks?**
> * Nesterov acceleration is an excellent strategy for improving the convergence rate in optimization; yet, we would like to highlight that it is complementary to our key contributions since it does not directly address the scalability of an algorithm on multicore systems. A great direction for future research would be to equip Async-RED with Nesterov acceleration (as was done in [2] for traditional asynchronous block-coordinate algorithms) to better understand the tradeoffs between momentum and scalability in multicore systems. On the other hand, the direct comparison between AGD-RED and Async-RED would be misleading since they correspond to different classes of algorithms (a close analogy would be the comparison between the vanilla SGD and vanilla AGD, with the former being faster on large-enough problems and the latter being faster on small-enough problems). We have added the discussion on Nesterov acceleration in the revision.
>
> **3. Secondly, Async-RED-SG seems to yield very limited acceleration in terms of convergence rate over the deterministic counterpart Async-RED-BG in CS example – what is the minibatch size used in this experiment? Would reducing the minibatch size yield a faster convergence rate? Meanwhile, the recovery result of Async-RED-SG is 1dB worse than the other methods, which seems a bit problematic (would a shrinking step-size help to fix this issue?).**
> * This observation highlights the tradeoff between the convergence rate and accuracy in stochastic optimization [3]. Our experiment runs Async-RED-SG with the minibatch 1120. A smaller minibatch would certainly reduce the per-iteration complexity; however, it would also reduce the accuracy of Async-RED-SG relative to GM-RED (see Theorem 2 and Fig. 3 (Left)). This trend is also illustrated in Table 2 (see additional_results.pad in the supplement) in terms of SNR. Note also how the error term due to stochastic processing in Theorem 2 is also proportional to the step size $\gamma$, which means that by using smaller $\gamma$, Async-RED-SG can approximate GM-RED as accurately as desired. However, a reduction in γ would also lead to slower convergence. One thus needs to tradeoff the desired accuracy against the desired speed to select a suitable configuration for Async-RED-SG. We have added a discussion of this in the revision.
>
> **4. It is highly suggested that the authors should also plot the convergence curves for the CT experiments as in the Fig.3.**
> * Prompted by your remark, we draw Figure 1 in additional_results.pdf. The figure plots SNR against the iteration number for Async-RED-BG/SG, Async-RED, and Gm-RED. Due to the lower per-iteration complexity, Async-RED-SG achieves the highest SNR value within the time budget of 1 hour.

---

> > ### Author Response · Authors · 2020-11-18
> > **Reference**
> >
> > **Reference:**
> >
> > [1] Z. Peng, T. Wu, Y. Xu, M. Yan, and W. Yin. Coordinate-friendly structures, algorithms and applications. Adv. Math. Sci. Appl., 1(1):57–119, April 2016.
> >
> > [2] R. Hannah, F. Feng, and W. Yin. A2BCD: Asynchronous acceleration with optimal complexity. In International Conference on Learning Representations, 2019.
> >
> > [3] L. Bottou and O. Bousquet. The tradeoffs of large scale learning. In Proc. Advances in Neural Information Processing Systems 20, pages 161–168, Vancouver, BC, Canada, December 3-6, 2007.

---

### Official Review · AnonReviewer1 · 2020-10-28
**Review of "Async-RED: A Provably Convergent Asynchronous Block Parallel Stochastic Method using Deep Denoising Priors"**

**Rating:** 6
**Confidence:** 2

**Review:**

Due to the growth of data sets in a lot of applications, it is important to develop algorithms to achieve great performance but with significantly reduced computational cost. The paper proposes asynchronous type of parallel algorithms by combining the pre-trained deep denoisers. In particular, batch gradient and stochastic gradient are applied to the proposed algorithm framework by taking advantage of the coordinate separable structures of the problem. Convergence of the algorithms are guaranteed under the four specified assumptions. Numerical experiments on the CT image reconstruction have justified the proposed efficiency and significant improvement in terms of running time. The importance and contribution of this work in compressive sensing algorithms stand. However, the novelty of the methods look incremental. There are some other issues listed as follows.

1. In p.1, is there any condition on the comparison, e.g., m<<n, required in the introduction?
2. The denoised version $D_\sigma(x^*)$ by some image denoiser essentially provides a more accurate estimate of $x^*$. Can this be replaced by other similar operators? Also, in the compressive sensing, the recovered image $x^*$ at the first few iterations may not be good enough, will the application of this operator make it worse? Does the parameter $\sigma$ need to tune or update dynamically in the iterations?
3. In Alg.1-2, the two operators read() and minibatch() should be explicitly defined.
4. In the numerical experiments, discussion on the influence of the block/minibatch size on the performance could be strengthened. In the ASYNC-RED-SG, how many trials were conducted to get an average performance? Robustness to the noise could be analyzed.

---

> ### Author Response · Authors · 2020-11-18
> **Response to Reviewer1**
>
> **1. In p.1, is there any condition on the comparison, e.g., m ≪ n, required in the introduction?**
> * Async-RED and its convergence analysis are independent of the specifics in the relationship between m and n. Compressive sensing (where indeed m ≪ n) is one of many possible applications of Async-RED. However, the method is also applicable to problems where m ≥ n, which has been mentioned in the updated manuscript.
>
> **2. The denoised version $D_\sigma(x^\ast)$ by some image denoiser essentially provides a more accurate estimate of $x^\ast$. Can this be replaced by other similar operators? Also, in the compressive sensing, the recovered image $x^\ast$ at the first few iterations may not be good enough, will the application of this operator make it worse? Does the parameter σ need to tune or update dynamically in the iterations?**
> * The revised manuscript mentions that Async-RED and its theoretical analysis are not restricted to denoising operators $D_\sigma$. Our theoretical analysis holds for any nonexpansive operator, which includes the traditional proximal operators (e.g., soft-thresholding) and the more recent artifact-removal operators [1,2].
> * The usage of the denoiser within iterations of our algorithm is similar to the usage of the soft-thresholding operator for l1-minimization in compressive sensing [3]. Thus, by using the operator in every iteration, Async-RED improves the solution compared to the scenario where such an operator is not used.
> * In the experiment section, we explicitly state that the parameter $\sigma$ is fixed across all iterations. Its value is fine-tuned for the best SNR performance (see details in Section 5), similarly to the way the regularization parameter is fine-tuned in traditional sparse recovery and compressive sensing.
>
> **3. There seems to be some mixup between BC-RED and GM-RED throughout. Are these different methods?**
> * Thanks for catching this. Indeed, BC-RED and GM-RED are different methods. This has been fixed in the updated manuscript.
>
> **4. In Alg.1-2, the two operators read(·) and minibatch(·) should be explicitly defined.**
> * We have explicitly defined the two operators in the updated manuscript.
>
> **5. In the numerical experiments, discussion on the influence of the block/minibatch size on the performance could be strengthened. In the Async-RED-SG, how many trials were conducted to get an average performance? Robustness to the noise could be analyzed.**
> * We performed additional experiments that have been included in the supplement. Table 1 (see additional_results.pdf in the supplement) summarizes the SNR values obtained for three minibatch sizes {60, 80, 120}. Async-RED achieves almost the same SNR values under these settings. Table 2 (see additional_results.pdf in the supplement) summarizes the SNR values for three minibatch sizes {1120, 2240, 3360}, which corresponds to 1/4, 1/2, and 3/4 of the full batch. As minibatch size increases, the final SNR performance improves, which is consistent with our theory. The revised manuscript explicitly mentions that the average performance in the manuscript is obtained by running a single trial for each image. We do not expect that running multiple trials would influence our results. We would like to highlight that our theoretical results stay valid for any amount of measurement noise. This is the reason we did not perform specific experiments on noise robustness, as they would not directly relate to our contributions. We include this discussion in the supplement.
>
> **Reference:**
>
> [1] K. Zhang, W. Zuo, and L. Zhang. Deep plug-and-play super-resolution for arbitrary blur kernels. In Proc. IEEE Conf. Computer Vision and Pattern Recognition (CVPR), pages 1671–1681, Long Beach, CA, USA, June 2019.
>
> [2] J. Liu, Y. Sun, C. Eldeniz, W. Gan, H. An, and U. S. Kamilov. Rare: Image reconstruction using deep priors learned without groundtruth. IEEE Journal of Selected Topics in Signal Processing, 14(6):1088–1099, 2020.
>
> [3] I. Daubechies, M. Defrise, and C. De Mol. An iterative thresholding algorithm for linear inverse problems with a sparsity constraint. Commun. Pure Appl. Math., 57(11):1413–1457, November 2004.

---

### Official Review · AnonReviewer3 · 2020-10-30
**Well explained method and interesting results**

**Rating:** 8
**Confidence:** 2

**Review:**

The paper describes a novel implementation of RED, regularization by denoising, which better leverages multicore architectures to achieve a significant speedup. The proposed implementation splits the gradient step into smaller components, which can each be executed independently on different cores and then used to update a shared copy. The crucial result is two sets of convergence guarantees showing that this delayed update will not cause too much error, even if the updates from different cores arrive at different times. The speedups achieved range from 6× to 8× on two tasks (compressive sensing and computed tomography reconstruction).

The paper is well organized and the details are explained well. The numerical results are convincing and the analysis is adequate. The main weak point of the paper is motivating the problem. In other words, it is not clear why a multicore method is needed, although the numerical results demonstrate this later on. For example, Section 3 starts by stating that “ASYNC-RED addresses the computational bottleneck…”, but what that computational bottleneck consists of is never explained. Figure 1 helps in explaining this, but much is left unexplained at this stage. Some discussion of the regime where multicore processing makes sense would also be in order. That being said, I think the results are interesting enough and the description of the method compelling enough that I recommend this work be published as part of the proceedings.

There are some small issues:
– On p. 2, H(x) is never defined.
– On p. 2, G is defined twice: once in eq. (2) and one in eq. (4). Presumably these refer to the same non-linear mapping.
– There seems to be some mixup between BC-RED and GM-RED throughout. Are these different methods?

---

> ### Author Response · Authors · 2020-11-18
> **Response to Reviewer3**
>
> **1. The main weak point of the paper is motivating the problem. In other words, it is not clear why a multicore method is needed, although the numerical results demonstrate this later on. For example, Section 3 starts by stating that “ASYNC-RED addresses the computational bottleneck. . . ”, but what that computational bottleneck consists of is never explained. Figure 1 helps in explaining this, but much is left unexplained at this stage. Some discussion of the regime where multicore processing makes sense would also be in order.**
> * We have clarified the importance of using a multicore system for imaging and rephrase the first sentence in Section 3 in the updated manuscript.
>
> **2. On p.2, $H(x)$ is not defined.**
> * The updated manuscript has defined $H(x)$ on p.2.
>
> **3. On p.2, $G(x)$ is defined twice: once in eq. (2) and one in eq. (4). Presumably these refer to the same non-linear mapping.**
> * Indeed, since they refer to the same mapping, we have deleted the definition of $G$ in eq. (4).
>
> **4. There seems to be some mixup between BC-RED and GM-RED throughout. Are these different
> methods?**
> * Thanks for catching this. Indeed, BC-RED and GM-RED are different methods. This has been fixed in the updated manuscript.

---

### Author Response · Authors · 2020-11-18
**General Response**

We thank the reviewers for their time and valuable feedback. We provide individual responses addressing the points of each review below, highlighting the changes we made in the revision to reflect these points. **We include an extra pdf file (additionl_results.pdf) in our supplement (.zip) for showing additional figures and tables.**

---

### Decision · Program_Chairs · 2021-01-07
**Final Decision**

**Decision:**

Accept (Spotlight)

**Comment:**

The reviewers agree that this paper overcomes a number of difficult algorithmic and technical challenges in parallelizing the RED method for image reconstruction.